# Development of a Novel, Potent, and Selective Sialyltransferase Inhibitor for Suppressing Cancer Metastasis

**DOI:** 10.3390/ijms25084283

**Published:** 2024-04-12

**Authors:** Han-En Tsai, Chia-Ling Chen, Tzu-Ting Chang, Chih-Wei Fu, Wei-Chia Chen, Ser John Lynon P. Perez, Pei-Wen Hsiao, Ming-Hong Tai, Wen-Shan Li

**Affiliations:** 1Institute of Chemistry, Academia Sinica, Taipei 115, Taiwansandisefu@gmail.com (C.-W.F.); spperez@gate.sinica.edu.tw (S.J.L.P.P.); 2Biomedical Translation Research Center, Academia Sinica, National Biotechnology Research Park, Taipei 115, Taiwan; 3Department of Chemistry, National Central University, Taoyuan 320, Taiwan; 4Department of Chemistry, National Taiwan Normal University, Taipei 106, Taiwan; 5Department of Applied Chemistry, National Yang Ming Chiao Tung University, Hsinchu 300, Taiwan; 6Sustainable Chemical Science and Technology, Taiwan International Graduate Program, National Yang Ming Chiao Tung University and Academia Sinica, Taipei 115, Taiwan; 7Agricultural Biotechnology Research Center, Academia Sinica, Taipei 115, Taiwan; 8Doctoral Degree Program in Marine Biotechnology, National Sun Yat-Sen University, Kaohsiung 804, Taiwan; 9Center for Neuroscience, National Sun Yat-Sen University, Kaohsiung 804, Taiwan; 10Institute of Biomedical Sciences, National Sun Yat-Sen University, Kaohsiung 804, Taiwan; 11Ph.D. Program in Drug Discovery and Development Industry, College of Pharmacy, Taipei Medical University, Taipei 110, Taiwan; 12Department of Medicinal and Applied Chemistry, College of Life Science, Kaohsiung Medical University, Kaohsiung 807, Taiwan

**Keywords:** sialyltransferase inhibitor, breast cancer, melanoma, metastasis, integrin sialylation

## Abstract

Sialyltransferase-catalyzed membrane protein and lipid glycosylation plays a vital role as one of the most abundant post-translational modifications and diversification reactions in eukaryotes. However, aberrant sialylation has been associated with cancer malignancy and metastasis. Sialyltransferases thus represent emerging targets for the development of small molecule cancer drugs. Herein, we report the inhibitory effects of a recently discovered lithocholic acid derivative FCW393 on sialyltransferase catalytic activity, integrin sialyation, cancer-associated signal transduction, MDA-MB-231 and B16F10 cell migration and invasion, and in in vivo studies, on tumor growth, metastasis, and angiogenesis. FCW393 showed effective and selective inhibition of the sialyltransferases ST6GAL1 (IC_50_ = 7.8 μM) and ST3GAL3 (IC_50_ = 9.45 μM) relative to ST3GAL1 (IC_50_ > 400 μM) and ST8SIA4 (IC_50_ > 100 μM). FCW393 reduced integrin sialylation in breast cancer and melanoma cells dose-dependently and downregulated proteins associated with the integrin-regulated FAK/paxillin and GEF/Rho/ROCK pathways, and with the VEGF-regulated Akt/NFκB/HIF-1α pathway. FCW393 inhibited cell migration (IC_50_ = 2.6 μM) and invasion in in vitro experiments, and in in vivo studies of tumor-bearing mice, FCW393 reduced tumor size, angiogenesis, and metastatic potential. Based on its demonstrated selectivity, cell permeability, relatively low cytotoxicity (IC_50_ = 55 μM), and high efficacy, FCW393 shows promising potential as a small molecule experimental tool compound and a lead for further development of a novel cancer therapeutic.

## 1. Introduction

Cell surface glycans provide a protective shield for the cell while mediating essential interactions with the extracellular matrix. Alterations in the glycosylation patterns of membrane lipids and proteins are associated with the progression of numerous diseases [1,2]. In cancer, aberrant glycosylation has been implicated in promoting tumor growth and metastasis [3,4]. A key characteristic of cancer cells is hypersialylation, which results from an imbalance in cellular sialidase and sialyltransferase activities [5,6]. Sialyltransferases (STs) catalyze the transfer of the sialyl group from the corresponding cytidine monophosphate (CMP)-conjugate to the glycan termini of proteoglycans and glycoproteins. Numerous studies have linked abnormally high levels of sialyltransferase activities to cancer [7]. Specific examples include the association of the sialyltransferases ST6GALNAC1 [8] and ST6GALNAC5 with breast cancer [9], ST3GAL6 with multiple myeloma and hepatocellular cancer [10,11], and ST6GAL1 with ovarian cancer [12,13]. In view of these findings, we have focused our work on the discovery of a cell-permeable and selective small molecule inhibitor of ST6GAL1 for use as an experimental tool compound and as a lead for the further development of a preclinical cancer drug candidate.

Known small-molecule ST inhibitors (recently reviewed in [14,15,16,17]) include sialic acid derivatives [18,19,20,21,22], CMP–sialic acid analogs [23], cytidine derivatives [24,25,26], oligosaccharides [27,28], aromatic compounds [29], flavonoids [30], lithocholic acid analogs [31,32,33,34,35,36], the fungal metabolite stachybotrydial [37], and polyoxometalates [38]. We have previously shown that the lithocholic acid derivative AL10 suppresses adhesion, migration, and invasion of human lung cancer cells and inhibits tumor metastasis in an animal model [33]. In addition, we found that another derivative, Lith-O-Asp, suppresses FAK/paxillin signaling, angiogenesis, and metastasis in lung cancer [32]. Despite their potency, AL10 and Lith-O-Asp both lack isozyme preference or selectivity toward different sialyltransferases, which are distinguished by the acceptor substrate targeted, the glycosidic linkage formed, and by the biochemical functions carried out in normal and cancerous cells.

In this regard, the present study was carried out with the aim of developing a novel lithocholic acid derivative that is both cell permeable and possesses selective N-glycan sialylation inhibition, and assessing its functional impact on cancer metastasis. Since integrins promote tumor metastasis and resistance during the course of chemotherapy [39,40,41,42,43], and because integrin-switching limits RNAi-based strategies [44,45], small molecule inhibitors of integrin sialylation remain necessary. 

As a huge milestone in the development of ST inhibitors, we have recently reported the discovery of the first-in-class selective inhibitors of N-glycan STs, with the compound FCW34 displaying the highest efficacy in vitro and in vivo [46]. In our attempt to further optimize the structure and improve the potency, we designed FCW393, a novel analogue characterized by a distinct amide linkage between the lithocholic acid backbone and the aspartic acid moiety (Scheme S1). We then hypothesize that the stronger amide linker may provide the ST inhibitor with greater cellular stability and better pharmacokinetic profile. For this purpose, we synthesized and biologically evaluated the lithocholic acid derivative, FCW393. Herein, we further report the inhibitory effects of FCW393 on sialyltransferase catalytic activity, integrin sialyation, cancer-promoting signal transduction pathways, protein expression levels, and lastly, cancer cell (MDA-MB-231 and B16F10) migration and invasion in vitro, and tumor size, metastasis, and angiogenesis in vivo.

## 2. Results

### 2.1. FCW393 Inhibits Integrin Sialylation and MDA-MB-231 Cell Migration

Human sialyltransferases form four evolutionarily and functionally distinct families: the ST3GAL (1–6), ST6GAL (1 and 2), ST6GALNAC (1–6), and ST8SIA (1–6) families [47]. To test FCW393 ST isozyme preference, we measured the IC_50_ for the in vitro inhibition of the N-linked glycan α-2,6-sialyltransferase ST6GAL1, α-2,3-sialyltransferase ST3GAL3, α-2,8-polysialyltransferase ST8SIA4, and the O-linked glycan α-2,3-sialyltransferase ST3GAL1. The respective IC_50_ values observed for ST6GAL1 and ST3GAL3 were 7.76 ± 0.06 μM and 9.45 ± 0.11 μM (Appendix A). Interestingly, no significant inhibitory activity was observed for ST8SIA4 (IC_50_ > 100 μM) and ST3GAL1 (IC_50_ > 400 μM) at the highest tested concentrations. These findings clearly indicate that, unlike our previous lithocholic acid-derived inhibitors, AL10 and Lith-O-Asp [32,33], FCW393 displays substantial isozyme preference in ST inhibition, and that it predominantly inhibits the integrin-regulating ST, ST6GAL1, with a selectivity index of more than fifty-fold against ST3GAL1. 

We further examined the effect of FCW393 (1–30 μM) on integrin isoform sialylation in human breast adenocarcinoma MDA-MB-231 cells by Western blot analysis of the immunoprecipitated α-2,3-sialylated and α-2,6-sialyated protein fractions. Our results show that FCW393 inhibited, in a dose-dependent manner, the α-2,3- and α-2,6-sialyation of integrin isoforms α_v_, β_1_, β_3_, β_4,_ and β_5_ in MDA-MB-231 cells (Appendix A). These results correlate with the in vitro ST inhibition of both ST3GAL3 and ST6GAL1. 

To assess FCW393 cytotoxicity, the viability of MDA-MB-231 cells treated with 1–80 μM FCW393 concentrations was measured using MTT assay. The results obtained, and reported in Figure 1A, define an IC_50_ of 55.3 ± 0.8 μM. To avoid unwanted repercussions of cytotoxicity in cell migration studies, we used FCW393 concentrations ≤ 30 μM, well below the determined IC_50_. As reported in Figure 1B, the results obtained using the wound healing assay show that following a 24 h incubation period with 10 or 30 μM FCW393, MDA-MB-231 cells exhibited considerably less migration than the DMSO-treated (control) cells. Furthermore, MDA-MB-231 cells, which were treated with 0–30 μM FCW393 and monitored using the transwell migration assay, displayed reduced migration activity in a dose-dependent manner (Figure 1C). Data fitting determined the IC_50_ to be 2.6 ± 0.1 μM for the inhibition of MDA-MB-231 cell migration (Figure 1C). 

Taken together, these results show that at concentrations lower than the IC_50_ for cytotoxicity, FCW393 inhibits ST6GAL1 activity selectively, integrin sialylation, and cancer cell migration. Subsequent experiments were then designed and performed to probe the functional impact of FCW393 on cancer-promoting signal transduction pathways.

### 2.2. FCW393 Suppresses Integrin-FAK-Paxillin and NFκB Signal Transduction Pathways in MDA-MB-231 Cells

Talin-mediated integrin activation is essential to the activation of the FAK-paxillin and NFκB signaling pathways, both of which play key roles in metastasis [48,49]. To explore the possible mechanisms underlying FCW393 inhibition of cell migration, we tested the effect of FCW393 on these pathways. Specifically, MDA-MB-231 cells were treated with 10 and 20 μM FCW393 or with DMSO (control) prior to pull-down and Western blot analysis. The results, reported in Figure 1D,E, show that the levels of *p*-talin-1, talin-1, integrin β_4_, integrin β_5_, integrin α_4_, *p*-FAK, and *p*-paxillin are notably reduced by FCW393. Likewise, FCW393 treatment reduced the expression levels of cellular *p*-IKKα/β, *p*-IκB, and *p*-NFκB (Figure 1F). Together, our results indicate that FCW393 downregulates the integrin/FAK/paxillin- and NFκB-mediated signal transduction, which we attribute as a consequence of FCW393 inhibition of integrin sialylation (Appendix A).

### 2.3. Antineoplastic Efficacy and Antimetastatic Activity of FCW393 in a Xenograft Model of Human Breast Cancer in MDA-MB-231 Cell-Bearing Mice

The effect of FCW393 on tumor growth and metastasis was studied by using nude mice injected with MDA-MB-231/Luc human breast cancer cells in the abdominal mammary gland area as the experimental platform. After the tumor volume reached 100 mm^3^, the subject mice were administered FCW393 (10 mg/kg body weight) or 40% PEG400 in normal saline (control) via intraperitoneal injection every third day. Tumor growth was monitored by using bioluminescence imaging. Representative images recorded at 8 w are reported in Figure 2A. The luciferase activities measured periodically over the 8 w period are reported in Figure 2B whereas the tumor volumes are provided in Figure 2C. These results reflect reduced tumor growth associated with the FCW393-treated mice compared to the control mice (*p* = 0.045). Moreover, the FCW393-treated mice did not experience significant loss of body weight.

The prevalence of metastatic cell colonies was determined by analysis of H & E-stained lung tissue. As reflected in the representative images shown in Figure 2D, fewer metastatic colonies were observed for the FCW393-treated mice than for the control mice. Overall, these findings show that FCW393 treatment reduces tumor growth and metastatic potential in a breast cancer cell mouse model.

### 2.4. FCW393 Inhibits B16F10 Cell Colonization and Invasiveness

Having demonstrated the FCW393-induced suppression of tumor growth in a breast cancer cell mouse model, we next investigated the effect of FCW393 on B16F10 mouse melanoma cell proliferation, colony formation, and invasion. First, FCW393 cytotoxicity was determined via MTT assay by measuring cell viability over a 48 h period for FCW393-treated (1–80 μM) vs. DMSO-treated (control) B16F10 cells. The data reported in Figure 3A were fitted to determine the IC_50_ = 57.8 ± 1.1 μM. Guided by this result, we carried out further in vitro analyses at FCW393 ≤ 30 μM.

We then measured colony formation in B16F10 cell cultures which were treated with 10, 20, and 30 μM FCW393, and observed a dose-dependent decrease in the number of colonies formed relative to the DMSO control (see Figure 3B). Afterwards, we used the Boyden chamber assay to assess the effect of FCW393 on matrix-penetration potential. The results given in Figure 3C show that FCW393 reduces cell invasion in a dose-dependent manner. Likewise, FCW393 evidently retards the matrix-penetrating capability in a dose-dependent manner (Figure 3C, lower panel). Such findings indicate that FCW393 can attenuate metastasis by inhibiting B16F10 cell colonization and tissue invasion.

### 2.5. FCW393-Inhibition of Signal Transduction Pathways in B16F10 Cells

Consistent with the results obtained using MDA-MB-231 cells in Section 2.1, FCW393 decreased the extent of α2,3- and α2,6-sialylation of integrin isoforms α_4_, α_v_, β_1_, β_3_, β_4_, and β_5_ in B16F10 cells (Appendix A). In Section 2.2, we previously reported that FCW393 treatment decreases the integrin/FAK/paxillin levels and NFκB signal transduction pathway proteins in MDA-MB-231 cells. For B16F10 cells, the signal pathways examined also included the VEGF and Rho/ROCK-dependent pathways. The results obtained, and presented in Figure 3D–G, show that FCW393 dose-dependently suppresses the levels of proteins, p-FAK (Tyr397), p-FAK (Tyr576/577), p-FAK (Tyr925), p-paxillin (Tyr118), p-IKKα/β (Ser176/180), IKKα, IKKβ, p-IκBα (Ser32), p-NFκB (Ser536), NFκB p65, p-VEGFR2, VEGF, p-Akt1/2/3, HIF-1α, VEGF, Rho A/B/C, ROCK I & II, and Rac I, functioning in each of these pathways. These data suggest that FCW393 inhibits B16F10 cell growth, motility, colonization, and invasiveness through multiple signaling networks involving integrin/FAK/paxillin, NFκB, VEGF, and Rho/ROCK signal transduction pathways.

To further understand whether FCW393 treatment inhibits B16F10 cell colonization and invasion partially via metastasis-associated signaling networks, we studied the biological event of epithelial to mesenchymal transition (EMT) where a non-motile epithelial cell changes to a mesenchymal phenotype with invasive abilities. The pilot study revealed that FCW393 treatment might elicit EMT reversal, by increasing E-cadherin while decreasing N-cadherin (EMT promoter) and Twist and Snail (mesenchymal cell markers) levels (Appendix A). However, FCW393 had no discernible effect on vimentin expression. These results suggest that FCW393 may inhibit the EMT process in B16F10 cells as a result of reduced sialylation of the integrin repertoire (Appendix A), which consequently attenuated both Akt activation and Rho/ROCK signaling (Figure 3F,G).

### 2.6. FCW393 Attenuates Tumor Growth and Delays Lung Metastasis in Mice Bearing Established B16F10 Melanoma

To investigate the effect of FCW393 on the behavior of melanoma cells in vivo, mice bearing established B16F10 cells were treated with FCW393 (10 mg/kg body weight) every other day for 26 d, during which time the tumor volume was periodically measured to monitor tumor progression. At day 26, the tumors of the FCW393-treated mice showed a significant reduction in size, which corresponds to 50% tumor growth inhibition based on quantification analysis (Figure 4A). The rate of tumor growth measured over the 26-day period was observed to be significantly slower for the FCW393-treated mice than for the control mice (Figure 4B).

After the mice were sacrificed, the melanoma tissues were harvested and examined by immunostaining (Appendix A). Immunohistochemical analysis was used to investigate *p*VEGFR2 (top) and VEGF (bottom) expression level in vivo. Reduction in VEGFR2 and VEGF expression levels in melanoma tissues was confirmed, indicating FCW393-mediated reduction in *p*VEGFR2 and VEGF expression in vivo. This finding is consistent with the above observation that FCW393 dose-dependently suppresses the levels of *p*VEGFR2 and VEGF proteins in B16F10 cells in vitro (Figure 3F). However, we do not exclude the possibility of either a FCW393-mediated direct effect on VEGFR sialylation (likewise for integrin) or an FCW393-associated gene expression effect on VEGFR signaling.

Next, to assess tumor angiogenesis, we examined the effect of FCW393 on the number and morphology of tumor blood vessels using an anti-CD31 antibody for immunohistological staining of tumor tissue harvested from the mice at day 26. The representative images measured for FCW393-treated vs. control mice, reported in Figure 4C, revealed that FCW393 treatment significantly decreased the CD31 vessel numbers in melanoma cells by more than 50% as compared to the control. Furthermore, by using cultured HUVECs, we evaluated the effect of FCW393 on various angiogenic processes in vitro. Results demonstrated that FCW393 substantially inhibited the migration and tube formation in HUVECs (Appendix A). Altogether, such findings support the fact that FCW393 inhibits angiogenesis in vitro and in vivo. To further evaluate the effect of FCW393 on tumor cell proliferation, tissue samples taken from day 26 mice were subjected to immunostaining using an anti-Ki67 antibody. The representative image, provided in Figure 4D, illustrates a smaller fraction of Ki67+ cells and therefore implies that less proliferation occurred in the FCW393-treated mice than in the control mice. 

In the B16F10 cell line study, the Rho/ROCK signal transduction pathway was confirmed to be associated with FCW393-mediated suppression of cancer cell growth, motility, colonization, and invasiveness (Figure 3G). Further investigation of the Rho/ROCK pathway was performed using tissue samples taken from day 26 mice (Appendix A). As the data in Appendix A show, expression levels of cyclin Rho A, Rho B, and ROCK II were observed to be significantly reduced upon treatment with FCW393 in six tissue samples (*n* = 6 in each group). This finding is in agreement with the observation that treatment of B16F10 cells with FCW393 induces suppression of the levels of Rho A/B and ROCK II proteins except Rho C, ROCK I, and Rac I. 

Lastly, we tested the effect of FCW393 on the metastatic potential of luciferase-expressing B16F10 cells (luc-B16F10 cells). Specifically, on day 0, the mice received intravenous injections of luc-B16F10 cells. On day 1, the mice were randomly divided into two groups which received either FCW393 (10 mg/kg body weight) or vehicle intraperitoneal injection every other day up to 14 d. The bioluminescence emissions of live FCW393-treated mice, and control mice, measured at day 7, are represented in Figure 4E. By comparison, the lungs of FCW393-treated mice showed 2.5-fold less bioluminescence than those of the control mice, indicative of a considerably lower level of resident luc-B16F10 cells. Similar results were obtained with mice evaluated at day 14 (see Figure 4F). The lungs of day 14 mice were harvested, homogenized, and assayed to reveal 81% less luciferase activity in the lungs of the FCW393-treated vs. the control mice (Figure 4G). Taken together, these results indicate that FCW393 suppresses B16F10 tropism to, and secondary cancer cell growth on, the lungs.

## 3. Discussion

Given the prior literature, it has been established that integrin function is affected and regulated by N-glycosylation since all integrins prominently carry N-glycans on the cell surface due to their having over 20 potential glycosylation sites [50,51]. Previous studies have linked the specific sialyltransferase, ST6GAL1, in the hypersialylation of β_1_ integrins which then leads to increased cell adhesion and migration in various cancers [52,53,54]. Such studies suggest that the inhibition of ST6GAL1 and diminished sialylation of integrins may be a practical and meaningful approach for cancer treatment. Hence, in this study, we first demonstrated that in vitro, FCW393 selectively inhibits the α-2,6-sialytransferase ST6GAL1 (IC_50_ = 7.8 μM) and the α-2,3-(*N*)-sialytransferase ST3GAL3 (IC_50_ = 9.45 μM), but not the α-2,3-(*O*)-sialytransferase ST3GAL1 (IC_50_ > 400 μM) and the α-2,8-polysialytransferase ST8SIA4 (IC_50_ > 100 μM). This unprecedented isozyme selectivity was not observed for our previously reported lithocholic acid-based ST inhibitors, Lith-O-Asp [32] and AL10 [33]. Building upon these observations, we showed that FCW393 inhibits integrin sialylation in breast cancer and melanoma cells, while suppressing cell proliferation, migration, and invasion. Administration of FCW393 to MDA-MB-231 tumor-bearing mice reduces tumor size and lung metastasis, whereas in vivo studies carried out with the melanoma mouse model showed that FCW393 suppresses tumor growth, metastasis, and angiogenesis. To further probe the mechanism of action of FCW393, we determined its effect on the signal transduction pathways associated with the focal adhesion complex and the VEGF receptor, as depicted in Figure 5.

The focal adhesion complex is a dynamic structure formed at the cytoplasmic face of the cell membrane, which mediates cell spreading, growth, and differentiation. The complex component, integrin interacts with proteins of the extracellular matrix and with cytoplasmic signal transduction adaptor proteins such as focal-adhesion kinase (FAK), integrin-linked kinase (ILK), vimentin, talin, and paxillin [55]. FAK-associated signaling controls cell motility and invasion [56,57] and regulates the lateral movement of integrin clusters in cancer [58]. By using FCW393-treated MDA-MB-231 and B16F10 cells, we were able to observe that FCW393 inhibits talin-1, FAK, and paxillin phosphorylation. We hypothesize that FCW393 inhibits cell motility and migration via the suppression of the integrin–FAK–paxillin signal transduction pathway. This finding is consistent with the observations described in our previous results [46].

VEGF is a key mediator of angiogenesis in cancer. The VEGF receptor signal transducers *p*-Akt and *p*-NFκB regulate HIF-1α and it, as a result, regulates the transcription of the VEGF gene (Figure 5). The IKK complex is the central regulator of NFκB activation. Previous studies have implicated sialyltransferases in the regulation of *p*-Akt and *p*-NFκB signaling [59,60,61]. In the present study, we used B16F10 cells to demonstrate that FCW393 reduces VEGF, p-VEGFR2, *p*-Akt, and HIF-1α levels and by using B16F10 and MDA-MB-231 cells, we showed that FCW393 reduces p-IKKα/β, p-IκBα, and p-NFκB levels. 

The regulation of cancer cell motility and invasion is reported to be associated with the Rho/Rho-associated protein kinases (ROCK) pathway. The Rho/Rho-associated coiled-coil containing protein kinase (ROCK) signal transduction pathway is also an integral part of VEGF-mediated angiogenesis and is not only implicated in VEGF signaling, but also involved in numerous processes necessary for angiogenesis to occur, including endothelial cell migration, survival, and cell permeability [62,63]. FCW393 reduced the levels of *p-*VEGFR2, VEGF, RhoA, RhoB, and ROCKII in the mouse B16F10 cell tumors as well as in cultured B16F10 cells. 

Based on these findings, we propose that FCW393 inhibits B16F10 and MDA-MB-231 cell metastasis and angiogenesis by inhibiting ST6GAL1-mediated surface protein sialylation, thereby suppressing the signal transduction pathways regulated by these proteins. With the advantages of FCW393 being cell permeability, displaying considerably low cytotoxicity, and having high potency, it may prove to be an effective small molecule experimental tool compound as well as a promising lead inhibitor in the development of a novel cancer therapeutic. 

## 4. Materials and Methods

### 4.1. Determination of FCW393 IC_50_ for In Vitro ST Inhibition

The recombinant Human ST6GAL1 (#7620-GT) and ST3GAL1 (#6905-GT) were purchased from R&D systems. All inhibition assays were carried out in duplicate. Reaction solutions initially contained ST6GAL1 (0.5 μg), Galβ1–4GlcNAc (25 μM), CMP-Neu5Ac (1 mM), and FCW393 (0, 1.25, 2.5, 5, 7.5, 10, 20 μM) in 50 μL of buffer (25 mM Tris buffer, 150 mM NaCl, 5 mM CaCl_2_, and 10 mM MnCl_2_) at 37 °C. Following incubation for 2 h, the reaction solutions were heated at 100 °C for 10 min and then separated by reversed-phase chromatography using NH_4_HCO_3_ (0.1M, pH 8.5) and methanol as eluent. The elution of the unconsumed Galβ1-4GlcNAc (retention time 30.4 min) and the sialylated product (retention time 29.4 min) were monitored at 348 nm. The kinetic data were obtained by fitting the initial rate data to the nonlinear regression equation. Inhibition of ST3GAL1 was tested in similar manner except that the assay mixtures contained the different components: MES buffer (200 mM), NaCl (100 mM), EDTA (0.5 mM), Triton X-100 (0.01%), ST3GAL1 (0.5 μg), T-antigen (2.5 mM), CMP-Neu5Ac (1 mM), and FCW393 (20 or 400 μM), which were mixed to a total volume of 50 µL. The assay mixture was incubated at 37 °C for 20 min and then quenched by heating up to 100 °C for 10 min. The sialylated product was resolved with reversed-phase HPLC at 310 nm.

The recombinant Human ST3GAL3 (#10554-GT) and ST8SIA4 (#7027-GT) were purchased from R&D systems. For the ST3GAL3, the activity assay was carried out according to the manufacturer’s protocol. The recombinant Human ST3GAL3 0.2 μg were incubated with CMP-sialic acid (Sigma-Aldrich, St. Louis, MI, USA, #C8271) 200 μM, N-acetyllactosamine (Sigma-Aldrich, #A7791) 2 mM, and coupling phosphatase2 0.1 μg supplied in EA002 kit (R&D systems, Minneapolis, MN, USA) at 37 °C for 20 min. The ST8SIA4 was tested in the similar manner except that the assay mixtures contained the different components: the recombinant Human ST8SIA4 0.25 μg were incubated with CMP-sialic acid (Sigma-Aldrich, #C8271) 0.25 mM, rhNCAM-1/CD56 (R&D systems, #2408-NC) 8 μg and coupling phosphatase2 0.1 μg supplied in EA002 kit (R&D systems) at 37 °C for 20 min. The kinetic data were obtained by fitting the initial rate data to the nonlinear regression equation.

### 4.2. Cell Culture

MDA-MB-231 human breast cancer cells and B16F10 mouse melanoma cells were obtained from the cell bank of the Bioresource Collection and Research Center, Hsinchu, Taiwan (BCRC#60425; BCRC#60031). The cells were maintained in complete medium made up of Dulbecco’s Modified Eagle Medium (DMEM; Gibco, Waltham, MA, USA) with 10% fetal bovine serum (FBS; Gibco), 2 mM glutamine, 100 mg/mL streptomycin, and 100 U/mL penicillin (Gibco), at 37 °C in 5% CO_2_ atmosphere.

### 4.3. Evaluation of FCW393 Cytotoxicity

Cells (~1 × 10^4^) were seeded into the wells of a 96-well plate that contained DMEM and 10% FBS. FCW393 was added to the wells in specified amounts. The plates were incubated for 48 h at 37 °C in 5% CO_2_ atmosphere, after which cell viability was evaluated using the reagent 3-(4,5-dimethylthiazol-2-yl)-2,5-diphenyl tetrazolium bromide (MTT, Sigma) according to the manufacturer’s specifications. For the control experiments, 0.1% DMSO was used in place of FCW393. The mean ± SEM was determined for 4 independent trials.

### 4.4. Wound Healing Assay

The effect of FCW393 on cell migration was assessed using a scratch wound assay as described previously [32]. Briefly, cells were added (in duplicate) to 6-well plates containing 10% FBS DMEM medium to form confluent monolayers. Following overnight incubation at 37 °C in 5% CO_2_ atmosphere, a scratch wound was applied using a sterile 100 μL plastic pipette tip. Debris was removed from the scratch by washing the cells with PBS. Specified amounts of FCW393 in 10% FBS were then added to the wells. The cells were allowed to migrate into the scratch for 24 h. The closure extent of the cell-free gap was detected by microscope with digital images system (Olympus; Tokyo, Japan) at different time intervals and measured by NIH image program.

### 4.5. Transwell Migration Assay

The cell migration assay was performed using transwell chambers equipped with a filter insert (8-μm pore; Falcon, BD Biosciences, Franklin Lakes, NJ, USA). Cells were seeded onto the transwell insert at a density of 8 × 10^4^ cells/mL of serum-free media, after which a specified amount of FCW393 added to the upper chamber. For the control assay, 0.1% DMSO alone was added. Complete serum DMEM medium was added to the lower chamber to serve as a chemoattractant. Following 16 h of incubation at 37 °C in 5% CO_2_ atmosphere, the invading cells were fixed by methanol and stained with 10% Giemsa solution (Sigma-Aldrich, St. Louis, MO, USA) for 30 min. The invaded cell number was counted from three different low-power fields. All experiments were carried out in triplicate.

### 4.6. Cell Invasion Assay

The cell invasion assay was carried out using a Boyden chamber as previously described [64]. First, a polycarbonate filter (8 μm pore size Nucleopore; Costar, Cambridge, MA, USA) was coated with Matrigel (1:2 dilution; BD Biosciences, San Jose, CA, USA). Next, 50 μL of 3 × 10^5^ B16F10 melanoma cells/mL in serum-free DMEM containing FCW393 (at the specified concentration), were added to the upper chamber while complete medium was added to the lower chamber. The Boyden chamber was placed in a humidified CO_2_ incubator at 37 °C for 16 h. Traversed cells were fixed by methanol treatment for 10 min and stained with 10% Giemsa solution (Sigma-Aldrich, St. Louis, MO, USA) for 20 min. The cell number was counted using three different low-power fields and expressed as the mean + SEM.

### 4.7. Western Blot Analysis

Protein samples (30 μg) were separated on 6–10% SDS-PAGE gels and then transferred to Immobilon-P membranes (Millipore Corp., Burlington, MA, USA) by electroblotting with electroblot transfer buffer. The proteins of interest were visualized using antibodies against talin-1, FAK, paxillin, IKKα, IKKβ, NFκB, IκBα, β-tubulin (Cell Signaling), phospho-talin, phospho-FAK (Tyr 397) (Tyr 576/577) (Tyr925), phospho-IKKα/β, phospho-NFκB, phospho-IκBα, phospho-paxillin (Cell Signaling), β-actin (BD Biosciences), and secondary antibodies anti-mouse IgG and anti-rabbit IgG (Perkin Elmer, Waltham, MA, USA) in conjunction with SuperSignal^®^ West Pico reagent (Thermo scientific, Waltham, MA, USA).

### 4.8. Lectin Affinity Assay Followed by IP-Western Blotting

The lectin affinity assay was carried out according to a published protocol [32]. Accordingly, cell surface α-2,3 and α-2,6-sialylated antigen-expressing proteins were specifically captured using biotinylated *Maackia amurensis* lectin II (MALII) and *Sambucus nigra* lectin (SNA) (Vector Laboratories, Inc., Newark, CA, USA), respectively. Approximately 500 μg of total cell lysate protein were incubated (with rotation) with MALII or SNA for 16 h at 4 °C. The lectin–protein complex was captured using streptavidin-conjugated agarose beads. The beads were washed three times with radioimmunoprecipitation assay (RIPA) buffer before eluting the bound proteins with SDS-PAGE sample buffer. The protein samples obtained were subjected to Western blot analysis using Integrin-β_1_, -β_3_, -β_4_, -β_5_, -α_4_, -α_5,_ and -α_v_ antibody (Cell Signaling, Danvers, MA, USA, Integrin antibody sampler kit, #4749).

### 4.9. Breast Cancer Mouse Model

Four- to five-week-old athymic nude mice (BALB/cAnN.Cg-*Foxn1^nu^*/CrlNarl obtained from National Laboratory Animal Center of Taipei, Taiwan) were raised in a controlled pathogen-free environment. Animal studies were approved by the Animal Core Facility of the Academia Sinica. MDA-MB-231/ Luc cells were kindly provided by Dr. Pei-wen Hsiao (ABRC Laboratory Animal Core Facility, Agricultural Biotechnology Research Center, Academia Sinica, Taipei, Taiwan). Cells were suspended in normal saline and then subcutaneously injected into the abdominal mammary gland area of nude mice. Body weight and tumor volume (=1/2 × length × (width)^2^) were measured weekly. At day 20 and a tumor volume of ~70–100 mm^3^, the mice were randomly divided into two groups. One group was treated with FCW393 at 10 mg/kg body weight, while the other group (control) was treated with vehicle (40% PEG400 + 60% normal saline). The growth and the spontaneous metastasis of the tumors were monitored by injection with Firefly D-Luciferin (Biosynth, Gardner, MA, USA) followed by imaging using an IVIS50 in vivo imaging system (Xenogen). The metastasized tumor tissues were dissected on day 56.

### 4.10. Metastatic Melanoma Mouse Model

Male C57BL/6JNarl (four- to six-week-old) mice were purchased from the National Laboratory Animal Center (Taipei, Taiwan) and housed under controlled pathogen-free conditions. All animal experiments were carried out using protocols approved by the Institutional Animal Care and Use Committee (IACUC) of National Sun Yat-Sen University (Kaohsiung, Taiwan; approval ID, 10510). B16F10 cells (5 × 10^5^) were resuspended in 0.1 mL phosphate-buffered saline (PBS) and injected into the tail vein of C57BL/6JNarl mice at day 0 to induce pulmonary metastasis. At day 1, mice were randomly divided into two groups. One group received 10 mg/kg body weight of FCW393 every two days until the mice were sacrificed, while the other group (control) received vehicle (40% PEG400 + 60% normal saline). Metastatic progression was monitored and quantified by using non-invasive bioluminescence as previously described [65]. The mice were sacrificed on day 14 at which time the lungs were harvested for histological analysis.

### 4.11. Histological Analysis

Following standard sectioning methods and staining with hematoxylin/eosin, mouse tissues were fixed for histological analysis by treatment with either 4% paraformaldehyde or Bouin’s solution for 24 h.

### 4.12. Immunohistochemistry

Paraffin-embedded tissue blocks were sectioned into 3 μm slices and mounted on poly-L-lysine-coated slides. After deparaffinization, the slides were blocked by treatment with 3% hydrogen peroxide for 10 min and subjected to antigen retrieval by heating in 10 mM citrate (pH 6.0) for 15 min in a microwave. Next, anti-CD31 (1:100 dilution; BD) and anti-Ki67 (1:200 dilution; Novocastra, Newcastle upon Tyne, UK) antibodies were applied onto the sections and incubated overnight at 4 °C. Following repeated washings with PBS, the sections were treated with horseradish peroxidase/Fab polymer conjugate (Polymer detection system, Zymed, South San Francisco, CA, USA) for 30 min. After rinsing with PBS, the sections were incubated with peroxidase substrate diaminobenzidine (1:20 dilution, Zymed) for 5 min. Thereafter, the sections were counterstained with Gill’s hematoxylin for 2 s, dehydrated with serial ethyl alcohol, cleared with xylene, and mounted for analysis.

### 4.13. Statistical Analysis

Differences between the groups were statistically evaluated using the unpaired Student’s *t*-test. The results are presented as the mean ± SEM. All *p*-values were two-tailed, and a *p*-value of less than 0.05 was considered to be statistically significant.

## Figures and Tables

**Figure 1 ijms-25-04283-f001:**
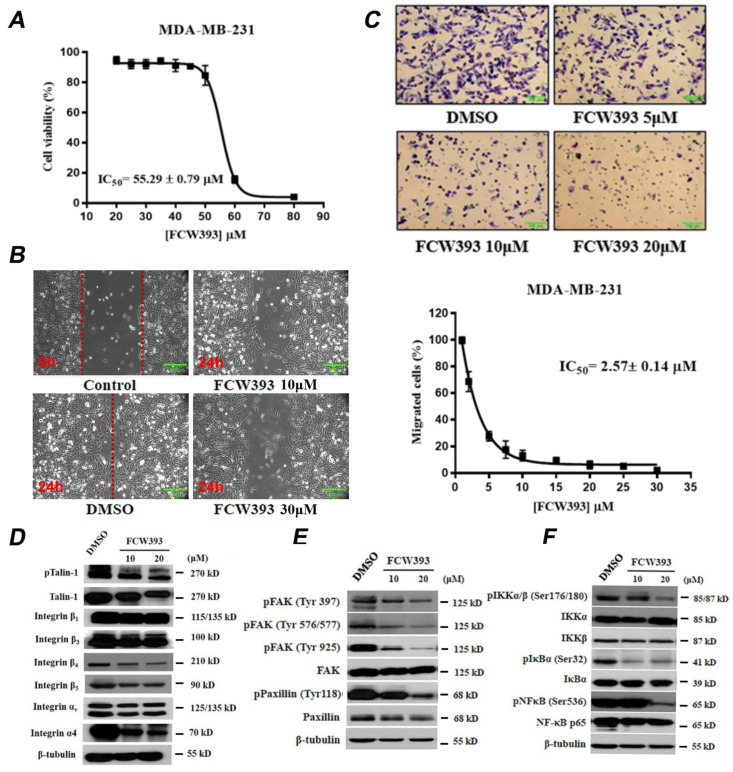
Analysis of cell viability, motility, and Western blot analysis in DMSO (control) and FCW393-treated MDA-MB-231 cells. (**A**) A plot showing the percentage of viable MDA-MB-231 cells remaining following 48 h incubation with 20–80 μM FCW393. Each data point represents the mean ± SEM of 4 independent measurements. (**B**) Representative photomicrographs of MDA-MB-231 cells treated with 10 or 30 μM FCW393, or with DMSO and subjected to the wound healing assay. Images were recorded at 0 and 24 h after wound induction; Scale bar: 200 μm. (**C**) *Top panel*: Representative photomicrographs of MDA-MB-231 cells treated with 5, 10, or 20 μM FCW393, or DMSO and subjected to the transwell migration assay); Scale bar: 100 μm. *Bottom panel*: A plot of the percentage of migrated cells vs. [FCW393] showing data fitting to define the IC_50_. Western blot analysis of integrin/FAK-paxillin/NF*κ*B signal transduction pathway proteins in MDA-MB-231 cells treated with (10 or 20 μM) FCW393, or DMSO (control) for 48 h. β-Tubulin was used as the loading control. The phosphorylated protein forms are represented by prefix “p” and the phosphorylation site is given in parentheses. (**D**) Western blots of phosphorylated and dephosphorylated talin, intergrins β_1–4,_ and integrins α_v_ and α_4_. (**E**) Western blots of FAK, *p*-FAK, paxillin, and *p*-paxillin. (**F**) Western blots of *p*-IKKα/β, IKKα, IKKβ, IκBα, *p*-IκBα, NFκB p65, and *p*-NFκB p65.

**Figure 2 ijms-25-04283-f002:**
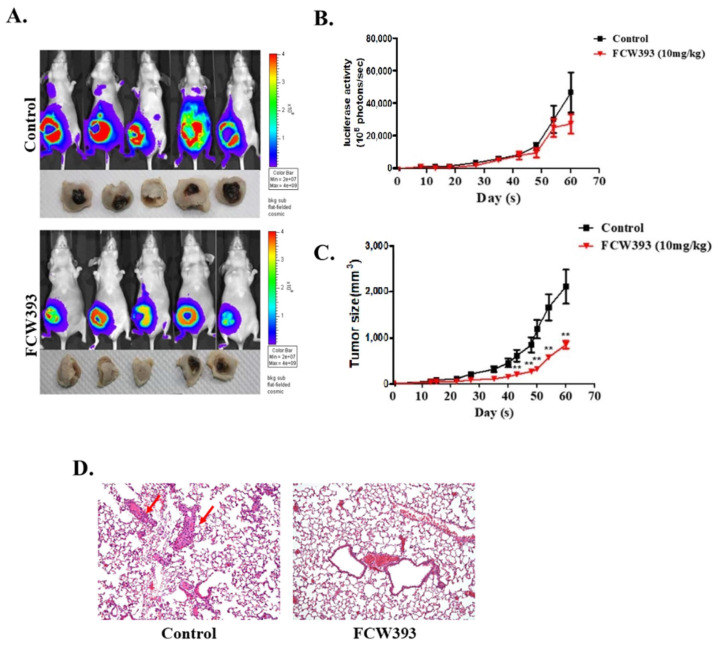
Analysis of tumor size and metastasis in nude mice bearing established MDA-MB-231/Luc tumors (100 mm^3^) and receiving 40% PEG400 + 60% normal saline (control) or FCW393 (10 mg/kg body weight) by intraperitoneal injection every other day. (**A**). In vivo luciferase bioluminescence images of tumors in mice measured at week 8 of FCW393 treatment. (**B**). Total tumor bioluminescence vs. treatment period (*n* = 10 per group). (**C**). Tumor size vs. treatment period (*n* = 9; ** *p* < 0.01). (**D**). Photomicrographs (100× magnification) of representative H & E-stained lung tumor tissue samples from control and FCW393-treated mice at day 60. The arrows point to the metastatic colonies in control group sample.

**Figure 3 ijms-25-04283-f003:**
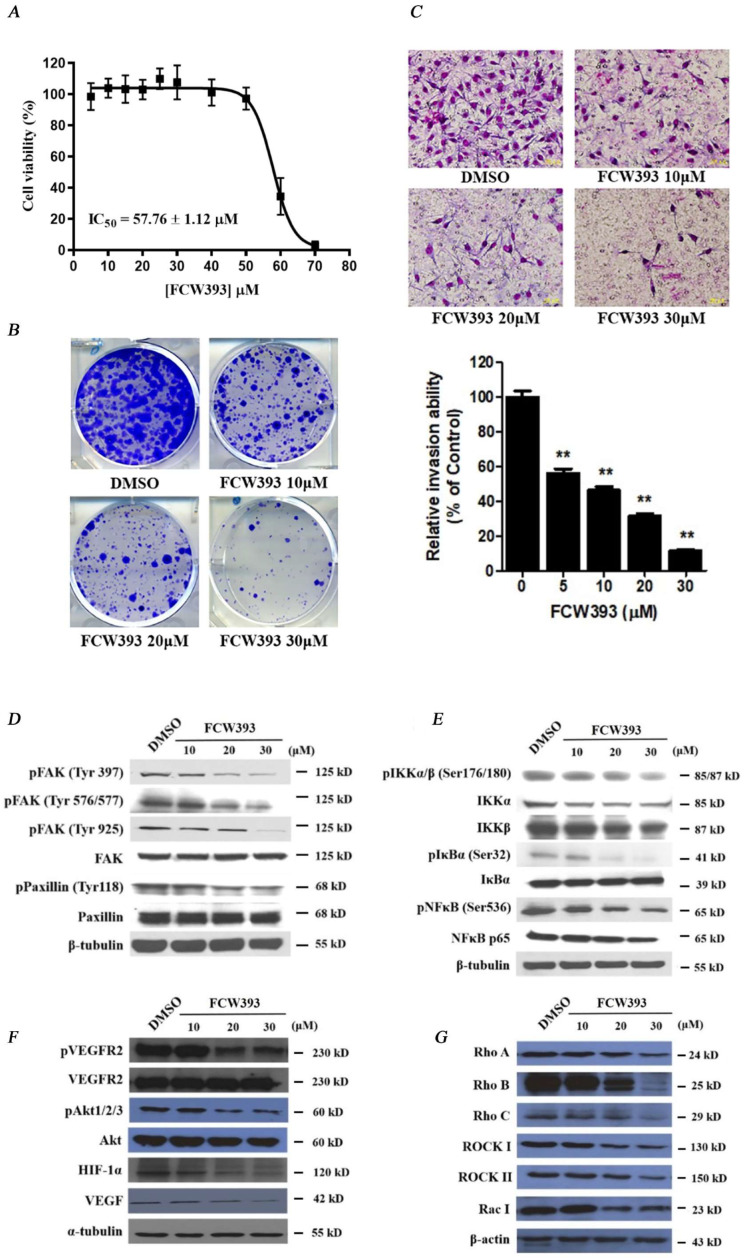
Analysis of cell viability, motility, and Western blot analysis in DMSO (control) and FCW393-treated B16F10 cells. (**A**) A plot showing the percentage of viable B16F10 cells remaining following 4 h incubation with 1–80 μM FCW393. Each data point is the mean ± SEM of 4 independent measurements. (**B**) Photographs of crystal violet-stained B16F10 cell colonies following incubation with 10, 20, or 30 μM FCW393 for 7 days. (**C**) *Top panel*: Representative photomicrographs of traversed B16F10 cells following treatment with 10, 20, or 30 μM FCW393 for 16 h, transmigration through a Matrigel-coated polycarbonate filter (8 μm pore size), fixing, and Giemsa staining); Scale bar: 100 μm. *Bottom panel*: Percentage of the total number of traversed FCW393 (0, 5, 10, 20, and 30 μM)-treated cells relative to the number of traversed control cells. All data are expressed as mean ± SEM based on 3 independent experiments (*** p* < 0.01). Western blot analysis of the integrin/GEF/Rho/ROCK and VEGF/Akt/NFκB/HIF-1α signal transduction pathway proteins in B16F10 cells treated with (10, 20, or 30 μM) FCW393 or DMSO (control) for 4 h. β-Tubulin or β-actin were used as the loading control. The phosphorylated protein forms are represented by the prefix “p” and the phosphorylation site is given in parentheses. (**D**) Western blot of *p*-FAK, FAK, *p*-paxillin, and paxillin. (**E**) Western blot of *p*-IKKα/β IKKα, IKKβ, IκBα, *p*-IκBα, NFκB p65, and *p*-NFκB p65. (**F**) Western blot of *p*-VEGFR2, VEGFR2, VEGF, *p*-Akt, Akt, and HIF-1α. (**G**) Western blot of Rho A-C, ROCK I & II, and Rac I.

**Figure 4 ijms-25-04283-f004:**
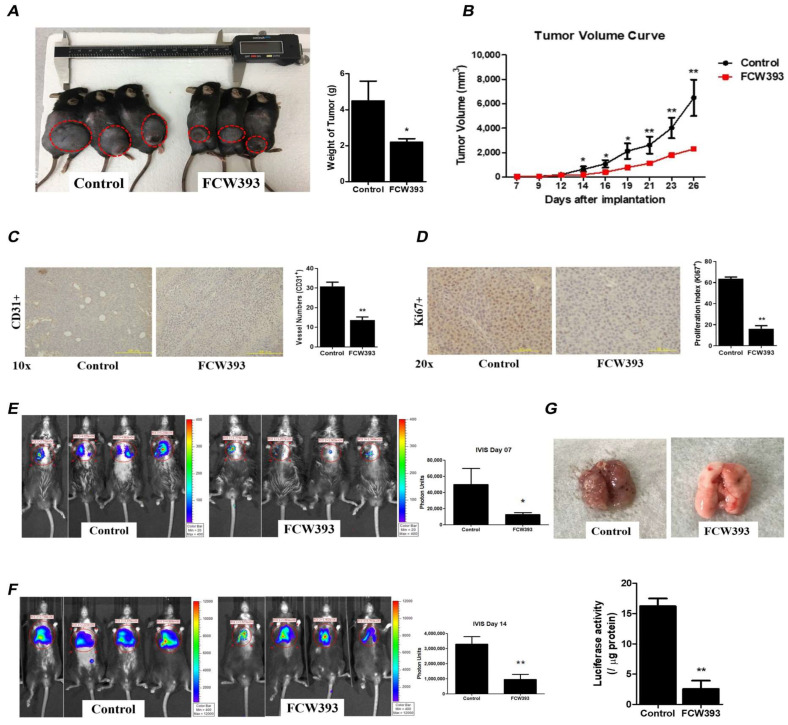
Analysis of tumor size, metastasis, and angiogenesis in C57BL/6JNarl mice bearing established B16F10 cell tumors (100 mm^3^) and starting on 10 d following implantation, receiving 40% PEG400 + 60% normal saline (control) or FCW393 (10 mg/kg body weight) by intraperitoneal injection every other day. (**A**) Photographs of FCW393-treated and control mice at day 26 following implantation. Tumors are located within the red circles; ** p* < 0.05. (**B**) A plot of (harvested) tumor volume vs. days following implantation and then FCW393 treatment (*n* = 10 in each group; ** p* < 0.05; ****, *p* < 0.01). (**C**) Photomicrographs (10× magnification) of fixed and CD31-immunostained tumor tissue samples at day 26 following initiation of FCW393 treatment (*n* = 5 in each group); Scale bar: 500 μm. (**D**) Photomicrographs (20× magnification) of fixed and Ki67-immunostained tumor tissue samples at day 26 following initiation of FCW393 treatment (*n* = 5 in each group); Scale bar: 200 μm. Determination of the effect of FCW393 on B16F10 cell metastasis. C57BL/6JNarl mice were intravenously administered with luciferase-expressing B16F10 cells at day 0 and starting on day 1, with an intraperitoneal injection of vehicle or FCW393 (10 mg/kg body weight) on alternate days. (**E**) Bioluminescence images determined at day 7. (**F**) Bioluminescence images determined at day 14. (**G**) Total bioluminescence determined for dissected and homogenized lungs. For (**E**–**G**), data values are reported as the mean ± SEM (*n* = 6 in each group; ** p* < 0.05; *** p* < 0.01).

**Figure 5 ijms-25-04283-f005:**
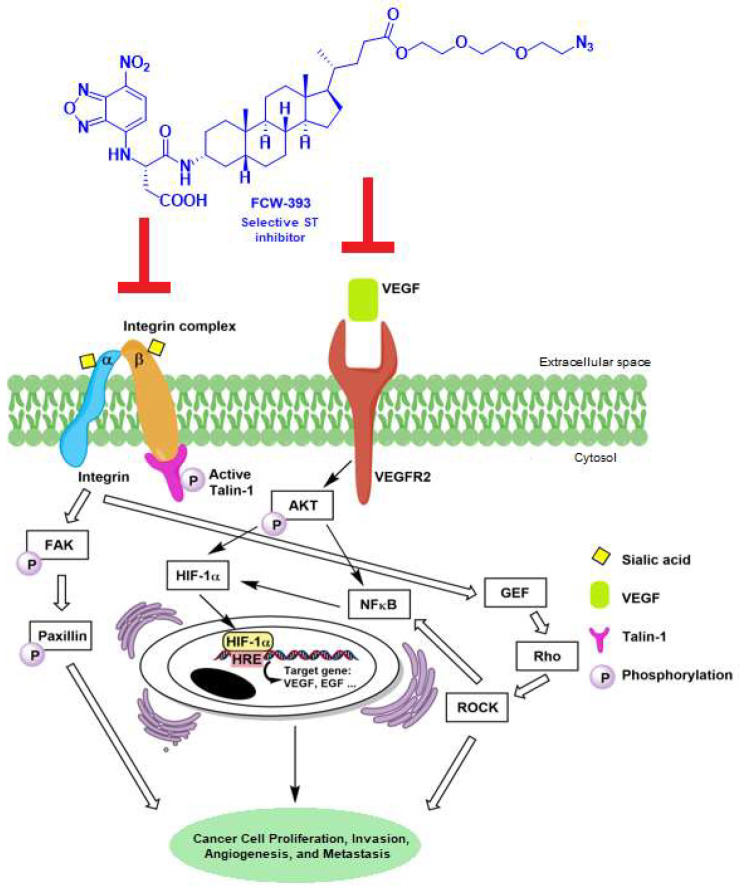
Proposed mechanism of FCW393-mediated suppression of tumor growth, inhibition of metastasis, and anti-angiogenesis.

## Data Availability

The data presented in this study are available on request from the corresponding author.

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
