# Peer review of "Development of a Novel, Potent, and Selective Sialyltransferase Inhibitor for Suppressing Cancer Metastasis"

_ijms, 2024, doi:10.3390/ijms25084283_

Round 1
Reviewer 1 Report (New Reviewer)
Comments and Suggestions for Authors
In this study, the authors have evaluated potential of FCW393 to inhibit B16F10 and MDA-MB-231 cell metastasis and angiogenesis by inhibiting ST6GAL1-mediated surface protein sialylation thereby suppressing the signal transduction pathways regulated by these proteins. The work deserves to be published after major revision:
1. What is FCW393? Why is it important? Why was it chosen? A brief introduction should be given.
2. Figure S1, Fig S5, Fig S6 are missing. Is the supplementary file different than the original blots file submitted?
3. Authors must provide loading controls for Fig S2 and Fig S3.
4. In Fig 1A, the authors say, ‘A plot showing the percent of viable MDA-MB-231 cells remaining following 48 h incubation with 20-80 μM FCW393.’ Please label the x-axis correspondingly. Same for other similar figures also.
5. Is it possible for authors to provide colored brightfield images for Giemsa staining in transwell assay (Fig 1C)?
6. In Fig S4, are the differences significant for N-cadherin, Twist and Snail?
7. Any specific reason why treatments were different for MDA-MB-231 and B16F10 tumor models? On page 5 line 161, FCW393 was injected every third day. While on page 8 line 245, authors mention injecting FCW393 every other day for 26d.
8. Magnified images of Fig 4C should be provided for better clarity of the results. Authors should also quantify the data.
9. Page 10 line 329, authors have mentioned that ‘In vivo studies carried out with breast cancer and melanoma mouse 329 models showed that FCW393 suppresses tumor growth, metastasis, and angiogenesis.’ However, angiogenesis assay was performed for melanoma model only. Kindly rephrase the conclusion.
Author Response
Please see the attached file.

Reviewer 2 Report (New Reviewer)
Comments and Suggestions for Authors
The authors of manuscript ijms-2922673 demonstrated that a novel selective Sialyltransferase inhibitor FCW393 can inhibit the in vitro migration and invasion of MDA-MB-231 breast cancer as well as B16F10 melanoma cells. Moreover, in vivo studies have shown that this small molecule inhibits the growth, metastasis and angiogenesis of both these tumors. They propose that FCW393 inhibits the metastasis and angiogenesis by inactivating the ST6GAL1 sialyltransferase, responsible for the sialylation of surface proteins. In this way, it suppresses signal transduction induced by them.
The data presented are robust and support the conclusions of the manuscript. The manuskrypt is well written and easy to read.
Small thing, in Fig 3 D, E, F, G is Fcw393 and in the rest of the text and Figs is FCW393.
Round 2
Reviewer 1 Report (New Reviewer)
Comments and Suggestions for Authors
The Authors have adequately answered all the concerns raised and hence, I recommend the manuscript to be accepted for publication.
This manuscript is a resubmission of an earlier submission. The following is a list of the peer review reports and author responses from that submission.
Round 1
Reviewer 1 Report
Comments and Suggestions for Authors
Why was the highest dose of FCW393 use in MDA-MB-231 and B16F10 cells in vitro?
How could the authors demonstrated FCW393 deliver to tumor sites?
Figure2D did not suggest that FCW393 inhibit tumor metastasis.
Please use flow cytometry to analyze the expression of intergrin on the cellular surface.
Reviewer 2 Report
Comments and Suggestions for Authors
In this manuscript entitled “Development of a Potent Sialyltransferase Inhibitor to Suppress Cancer Metastasis”, the authors Tsai et al. aimed to propose novel inhibitor (FCW393) of sialyltransferases deriving from lithocholic acid and having enzyme preference (or selectivity). Despite the important biological role of sialyltransferases, no specific inhibitor is currently known. However, this study suffers from serious flaws.
The inhibitory effect of this compound was studied in vitro only for two sialyltransferases ST3Gal I and ST6Gal I and not in the same conditions of reaction (Tris or MES buffer, detergent, EDTA, time of incubation?). These discrepancies should be at least explained. The origin of the enzymes used in this study should be given: are they truncated recombinant commercially available enzymes? The effect of FCW393 should be tested on closely related-sialyltransferases (ST3Gal II for instance) and at least on one enzyme of the different group of enzymes (ST6GalNAc, ST8Sia). These data with IC50 determination should be presented in the main text.
According to these first in vitro data, the FCW393 inhibitor would have a selective effect on α2,6-sialylation of N-glycans conferred by ST6Gal I. Yet, the authors use biotinylated Maackia amurensis lectin II (MAL-II) for their Lectin affinity assays followed by IP-western blotting. This lectin recognizes primarily α2,3-sialylated antigens of O-glycosylproteins. With this strategy, the authors show inhibition of various integrin (β1-, β3-, β4-, β5-, α4- and αV-integrin) sialylation by FCW393 in various cells MDA-MB-231 and B16F10) in culture. These partial data do not corroborate the in vitro assay of ST3Gal I and ST6Gal I. The Sambucus nigra lectin (SNA) should be used to follow α2,6-sialylation of N-glycans. The status of sialylated glycans of the various integrin (β1-, β3-, β4-, β5-, α4- and αV-integrin) should be given. Do they have sialylated N-glycans and/or O-glycans in the MDA-MB-231 and B16F10 cells?